# Severe Intentional Corrosive (Nitric Acid) Acute Poisoning: A Case Report and Literature Review

**DOI:** 10.3390/jpm13060987

**Published:** 2023-06-13

**Authors:** Alexandra Stoica, Cătălina Lionte, Mădălina Maria Palaghia, Irina Gîrleanu, Victoriţa Şorodoc, Alexandr Ceasovschih, Oana Sîrbu, Raluca Ecaterina Haliga, Cristina Bologa, Ovidiu Rusalim Petriş, Vlad Nuţu, Ana Maria Trofin, Gheorghe G. Bălan, Andreea Nicoleta Catana, Adorata Elena Coman, Mihai Constantin, Gabriela Puha, Bianca Codrina Morăraşu, Laurenţiu Şorodoc

**Affiliations:** 1Internal Medicine Department, Faculty of Medicine, “Grigore T. Popa” University of Medicine and Pharmacy, 700115 Iași, Romania; 2Second Internal Medicine Clinic, Sf. Spiridon Clinical Emergency Hospital, 700111 Iasi, Romania; 3General Surgery Department, Faculty of Medicine, “Grigore T. Popa” University of Medicine and Pharmacy, 700115 Iași, Romania; 4First General Surgery Clinic, Sf. Spiridon Clinical Emergency Hospital, 700111 Iasi, Romania; 5Gastroenterology Clinic, Sf. Spiridon Clinical Emergency Hospital, 700111 Iasi, Romania; 6Nursing Department, Sf. Spiridon Clinical Emergency Hospital, 700111 Iasi, Romania; 7Second General Surgery Clinic, Sf. Spiridon Clinical Emergency Hospital, 700111 Iasi, Romania; 8Infectious Diseases Department, Sf. Spiridon Clinical Emergency Hospital, 700111 Iasi, Romania; 9Preventive Medicine and Interdisciplinary Team Department, Sf. Spiridon Clinical Emergency Hospital, 700111 Iasi, Romania

**Keywords:** caustic, nitric acid, poisoning, attempted suicide, gastrointestinal endoscopy, esophageal stenosis

## Abstract

Despite being one of the most debilitating conditions encountered in the field of toxicology, there is a lack of neutralization measures for the toxins involved in acute corrosive poisoning, and this promotes progressive contact injury of deep tissues after poisoning has occurred. Multiple controversies still surround management strategies during the acute phase of poisoning and the long-term follow-up of the patient. Here, we report a severe case of intentional poisoning with nitric acid complicated by extensive injury of the upper digestive tract, multiple stricture formation, and complete dysphagia. Serial endoscopic dilation and insertion of a jejunostomy feeding tube were necessary, and underlying psychiatric illness negatively affected the outcome of the patient. We conclude that an interdisciplinary approach is necessary to properly reduce the extent of lesions and sequelae induced by corrosion. Early endoscopic mapping of injuries is of major importance to better predict the evolution and possible complications of poisoning. Interventional and reconstructive surgical procedures may significantly improve the life expectancy and quality of life of patients following intoxication with corrosive substances.

## 1. Introduction

Nitric acid (HNO_3_, also known as aqua fortis) is a highly corrosive mineral acid [1], and its acute toxicity is strongly connected to the route of exposure (inhalation, ingestion, dermal, or ocular). Since nitric acid is mainly used in industrial environments (dissolving noble metals, production of ammonium nitrate fertilizers, and manufacturing explosives), acute poisoning is most likely to occur in an occupational setting [2,3].

Nevertheless, commercial formulations of the compound are widely accessible to the general population and are intended for household use. Most available products contain approximately 56–68% nitric acid, and solutions containing 86–95% nitric acid are referred to as fuming nitric acid [1,4,5,6].

Although considered rare in the large spectrum of potentially lethal acute intoxications, corrosive-substance poisoning represents a major preventable public health problem [7,8,9,10,11,12,13,14]. Life-threatening injuries may develop within hours after oral exposure to corrosive substances [15]. Accidental poisonings, usually involving small amounts of the toxin, occur most often in children (68–80% of cases worldwide), while most intentional ingestions occur in adolescents and adults, and all of the cases represent a significant source of morbidity and mortality [16,17,18,19,20]. Indeed, the overall mortality rates after corrosive ingestion are reported to be as high as 20% [21]; however, in severe cases of corrosive-induced injuries (especially after suicide attempts), mortality can reach 75% [17].

The initial assessment of the patient in emergency settings must provide answers related to the need for intensive care unit (ICU) admission, risk factor assessment for the development of gastro-esophageal strictures, and the need for emergency or elective (reparatory) surgery [22]. The current recommendations and algorithms, although based on a low level of evidence, have important clinical and functional implications [23]. Extensive efforts must be made in the acute phase to improve short-term survival by limiting the extent of injuries and preserving the structural integrity of the digestive tube and surrounding organs [24].

Corrosive poisoning can induce short-term (esophageal and gastric perforation, mediastinitis, pneumomediastinum, and aspiration pneumonia) and long-term (gastro-esophageal strictures/gastric outlet obstruction, mediastinitis, esophago-respiratory fistula, malnutrition, and malignancy) complications, and proper management in both situations is essential [5,25].

Here, we report a severe case of a patient who attempted suicide by ingesting nitric acid, which was complicated by extensive gastro-esophageal necrosis and complete dysphagia secondary to multiple stricture formation treated with serial endoscopic dilation. The refusal of the patient to proceed with surgical treatment, as well as noncompliance with the instructions received from physicians (due to the patient’s decompensated psychiatric condition), contributed to the fatal outcome.

Although severe respiratory effects and/or lethality can occur after the inhalation of gases and vapors originating from nitric acid, this article refers exclusively to the acute toxicity associated with the ingestion of corrosive substances, such as nitric acid [26,27].

## 2. Case Presentation

A 46-year-old male was admitted to the emergency department 3 h after the intentional ingestion of an unspecified amount of nitric acid. As part of the Roma ethnic group, the patient worked as a coppersmith, handling concentrated solutions of nitric acid, which he used for the surface cleaning of copper vessels.

The patient had a medical history of depression (a bereavement reaction to his wife’s death 6 weeks previously) and chronic alcoholism. Suicidal ideation and deliberate self-poisoning were enhanced by ethanol co-ingestion.

The initial clinical manifestations included dysphagia, retrosternal and epigastric pain, vomiting, and productive cough. The clinical examination showed anxiety and agitation, a Coma Glasgow Scale score of 12, breath alcohol odor, blood pressure of 176/124 mmHg, a heart rate 84/minute, a respiratory of rate 21/min, oxygen saturation (SpO_2_) of 97% in room air, and absence of fever. Two episodes of hematemesis were witnessed in the emergency department.

The biological assessment confirmed the co-ingestion of ethanol, increased pancreatic enzymes suggestive of toxic acute pancreatitis, systemic inflammatory reaction (leukocytosis at 20,200/mm^3^ with neutrophilia of 90% and increased high-sensitivity C-reactive protein), thrombocytopenia, and increased serum lactate. The otorhinolaryngology evaluation showed mild congestion of the oropharyngeal and laryngeal mucosa, without signs of acute respiratory failure.

The emergency esophagogastroduodenoscopy revealed extensive esophageal injuries with multiple ulcerations, necrotic tissue, circumferential exudative lesions, and friability of the mucosa, as well as deep ulcerations with black discoloration in the stomach without signs of active bleeding (Figure 1).

The combined contrast-enhanced thoracic and abdominal computed tomography revealed parietal thickening involving the lower esophagus with superjacent dilation up to a caliber of 22 mm, a stomach and duodenum with thickened walls, and peri-duodenal fluid. No signs of pneumomediastinum or pneumoperitoneum were present.

The therapeutic sequences in this case included, foremost, the stabilization of the patient (endotracheal intubation and mechanical ventilation in the first 24 h, as well as hemostatic treatment) with careful monitoring in the intensive care unit and then admission to the general surgery unit for surveillance and follow-up in case of immediate complications (such as active hemorrhage or perforation). Complete restriction of oral intake, high-dose proton pomp inhibitors, and parenteral nutrition were initiated. Empiric antibiotic treatment (amoxicillin/clavulanic acid and clindamycin) was administered due to clinical, biological, and imagistic suspicion of tracheobronchial aspiration.

On the seventh day after ingestion of the nitric acid, the patient tested positive for SARS-CoV-2 (RT-PCR) and, due to the epidemiological context and the stability of the corrosive lesions, he was transferred to the Internal Medicine Department for the treatment of pneumonia.

Eleven days after the ingestion, the patient restarted oral feeding by ingesting small amounts of fluids, which was followed by the persistence of dysphagia and an accentuation of epigastric pain. Although parenteral nutrition was recommended, the patient disregarded the indications and continued to feed himself orally with poor digestive tolerance (nausea, vomiting, and upper abdominal pain). On the 13th day after the ingestion, the patient presented hematemesis and melena. An endoscopic evaluation was attempted, but a lack of cooperation on the part of the patient restricted the further continuation of the procedure due to safety reasons. Hemostatic treatment was administered, and remission of the bleeding signs was noted. In the following days, the patient’s clinical state permitted a new attempt to restart oral nutrition (liquids and semi-solid foods) with a favorable clinical outcome, and the patient was discharged on the 22nd day of the evolution of post-corrosive ingestion. Recommendations for gastroenterology evaluation (by esophagogastroduodenoscopy or by barium esophagogram) in an ambulatory setting were made to accurately assess the residual post-corrosive injuries (4 weeks after the exposure to nitric acid).

The long-term follow-up revealed the development of multiple esophageal strictures, with dysphagia for liquids and, intermittently, solids. The patient was non-compliant with the indications of the gastroenterologist and only appeared for two of the endoscopic esophageal dilation sessions with a 10–12 mm Controlled Radial Expansion balloon, mentioning that he could feed himself with semi-solid foods at home.

Three months after the ingestion, he presented himself again to our hospital for complete dysphagia for liquids. He refused a new endoscopic dilation session, which is why a feeding jejunostomy was inserted.

About 5 months after the insertion of the jejunostomy tube, the patient presented himself to our hospital again due to accidental self-suppression of the jejunostomy tube. At this moment, the patient showed a noticeable decrease in weight. Esophageal stripping with esophagoplasty was proposed to him, but he refused the surgical intervention, and a new insertion of the feeding jejunostomy was jointly decided by the medical team.

Approximately 10 months after the ingestion event, the patient arrived at the gastroenterology department again, where a new session of endoscopic esophageal dilation was performed, which was complicated by esophageal perforation, pneumomediastinum, minimal right pneumothorax, and right pleurisy. The patient continued to refuse surgical treatment for esophageal stenosis and pleurotomy. His general condition worsened, necessitating admission to the intensive care unit, orotracheal intubation, and mechanical ventilation. During this time, a medical committee consisting of an intensive-care physician, a thoracic surgeon, and a general surgeon was assembled, and it was decided, for the patient’s well-being, to perform a bilateral pleurotomy and drain the pleurisy. Considering his improved condition, the patient was extubated, with the resumption of respiratory function in normal parameters being achieved. Despite psychological counseling and psychiatric evaluation, the patient behaved aggressively, refused the administration of medication, and once again self-suppressed his jejunostomy tube.

The patient was supported throughout the hospitalization with parenteral nutrition and was discharged at the request of his family to be cared for at home. The subsequent follow-up (through his general practitioner) confirmed the death of the patient.

## 3. Discussion

An extensive search of the literature was performed with the journal search engines of Thompson ISI—Web of Science and PubMed. We used the MeSH terms “suicide”; “nitric acid”; “corrosive”; “caustic”; and “toxicity”, “intoxication”, or “poisoning” in different permutations. The titles and abstracts were initially screened, and relevant studies were selected for full-text evaluations. Additionally, we examined the citations of all resulting articles for any additional relevant references. Given the large distribution of pediatric cases, attentive selection of articles involving this category of patients was carried out to properly extract relevant information for the adult population.

### 3.1. Epidemiology

The epidemiological burden of acute corrosive poisoning is difficult to estimate given the inhomogeneous distribution of cases worldwide, corroborated by an increased number of undetected and unreported cases. An estimated 5000 to 15,000 new cases of caustic ingestions occur annually in the United States [28,29]. Prevention measures, such as population education and guidance; regulations involving the packaging of potentially corrosive substances; and limitations of free commercialization for the most potent agents, are all of major importance in reducing the incidence of corrosive injuries, especially when involving the pediatric population [20,30].

### 3.2. Pathophysiology

The acute toxicity of nitric acid is mainly associated with the extremely corrosive nature of this strong acid. Several factors can reflect the potential toxicity of the corrosive substances and can strongly interfere with the prognosis of the patient, such as the type (acidic or alkaline agents), the pH level (<2 or >12), the volume, the concentration, the time of contact, the state of presentation (liquid substances produce more extensive injuries than solids), and the intentionality of the poisoning (suicide attempt/accidental) [5,16,21,31].

Acidic or alkaline corrosive substances also represent a significant variable in determining the anatomical injury site. Acids will mainly affect the lower esophagus and the stomach, while alkaline chemicals will affect the oropharynx and the esophagus. One possible explanation for this could be the partial neutralizing effect that the endogenous gastric secretion might have when exposed to alkali substances [5,17,30].

Furthermore, repeated episodes of vomiting and/or the repeated passage of the corrosive substance in the first 6 h after ingestion induced by dyskinetic movements of the upper digestive tract may lead to tracheobronchial aspiration, thus further enhancing the potential of toxicity in the acute phase [5].

The ingestion of acidic substances induces acute coagulative necrosis (denaturation of all tissue proteins), accompanied by the formation of protective eschar, which explains the delayed risk of perforation that may occur, starting from day 4 [5,16,32]. On the other hand, alkali substances result in liquefactive necrosis with saponification of the fatty acids in cell membranes and thus tend to penetrate the gastro-esophageal structures deeper, generating microvascular thrombosis and an amplified inflammatory response [5,33,34,35]. Systemic effects may also occur, such as disseminated intravascular coagulation, multi-organ system failure, and sepsis [19].

### 3.3. Clinical Diagnosis

Due to its excellent oxidizing action, nitric acid induces an immediate reaction with any tissue, causing early symptoms such as oropharyngeal pain, dysphagia, excessive salivation, and upper gastrointestinal bleeding [27,36,37].

Although the clinical diagnosis may become evident in the presence of the patient’s relevant history, the most severe forms of intoxication tend to exhibit a very complex picture. Involvement of respiratory structures may lead to laryngospasm, stridor, dyspnea, hemoptysis, dysphonia, or respiratory distress signs [38,39,40,41,42].

In the case of esophageal perforation, the location of the pain will indicate the site of perforation in 70–90% of cases. Subcutaneous emphysema may be present when perforation of the cervical esophagus occurs, while hemodynamic deterioration, fever, or signs of multiorgan failure should raise suspicion of infectious complications and septic shock [43,44].

Even in the absence of these symptoms, potential harm may be considered, indicating the need to still perform endoscopy/bronchoscopy in selected cases (children, deliberate self-poisoning, and altered mental status). Another misleading element in proper clinical assessment after exposure to corrosive substances may be the different distribution of damaged tissues. A lack of obvious oropharyngeal lesions does not preclude the existence of distal injuries at multiple levels, and nor should it minimize the severity of the poisoning [36,45].

Moreover, the intensity of the primary complaints may vary according to the different phases of tissue remodeling (acute injury leading to necrosis, and then necrosis leading to granulation). Thus, no correlation can be established between the severity of the symptoms and the extent of the injuries [46]. The severity of dysphagia is mainly influenced by coexisting esophagitis rather than the degree of stenosis [47,48].

### 3.4. Primary Management

Estimations of the volume of the ingested substance by clinicians seem to be inaccurate, thus making it difficult to estimate the severity of intoxication using this criterion alone. 

Pre-hospital management should focus on establishing the toxicological context, identifying risk factors (extreme ages, such as children and the elderly; pregnancy; and comorbidities) and initiating supportive care measures (preventing vomiting and tracheobronchial aspiration by using antiemetics; 45° elevated positioning of the thorax; and avoiding epuration techniques, such as gastric lavage or induced emesis). Attempts to neutralize acids or alkalis are prohibited since exothermic reactions may occur, increasing the damage [23].

### 3.5. Paraclinical Evaluation and In-Hospital Management

Initial management should carefully assess the extent of injuries. Esophagogastroduodenoscopy remains the gold standard for diagnosis evaluation of corrosive-induced injures of the upper digestive tract. Furthermore, the precise endoscopic mapping of the injuries (location, extent, and severity) represents the most important determinant regarding the prognosis and treatment of the patient (Table 1).

Emergency endoscopy must be performed within the first 48 h after exposure to the toxin, with the optimal interval generally considered to be 12–24 h [5,36,49,50]. According to Zargar et al., delayed endoscopy (48–96 h after ingestion of corrosive substances) remains safe, and the subacute phase (5 to 15 days after corrosive ingestion) is, in fact, the moment most prone to perforation [49]. Relative contraindications to perform endoscopy include airway compromise, hemodynamic instability, and/or suspicion of perforation, requiring initial cardio–respiratory stabilization and accurate imagistic exclusion of esophageal and gastroduodenal perforations.

Since various gradings of the injuries can be highlighted throughout the length of the upper digestive tract [49], great benefits can result from an early complete endoscopic assessment mainly concerning the interventional and/or surgical therapeutical approach. Co-existing gastric and intestinal injuries can be found in about 20 to 62.5% of cases [51,52].

Recent research has involved a series of debates regarding the diagnostic and prognostic importance of computed tomography (CT) imaging as a non-invasive method in the assessment of corrosive-induced lesions [37,38,53,54,55]. Controversies regarding the necessity of performing routine endoscopy in all children exposed to toxins, especially in the absence of symptoms or visible oropharyngeal lesions, have arisen [44]. According to several studies, endoscopic evaluation is still superior in the first stages of the disease compared to computed tomography in estimating injury severity [56]. Nevertheless, Ryu et al. proposed a CT grading system (assessing the degree of esophageal damage and adjacent tissue) that was significantly correlated with the development of esophageal stricture [57].

In current practice, endoscopic and imagistic modalities offer complementary information and, in the absence of contraindications, sustain a holistic approach for the acutely poisoned patient. The CT scan is the investigation method of choice when perforation is suspected or when involvement of the mediastinal/abdominal structures needs to be assessed [37,58].

According to our experience, biological inflammatory markers (high-sensitivity C-reactive protein and white blood cell count) are significantly higher in patients with corrosive poisoning compared to other forms of acute intoxications (street drugs, toxic alcohols, and pesticides). There is pathogenic support for this since tissular injuries associated with corrosive exposure involve a sustained systemic inflammatory response that, nonetheless, must be differentiated from other possible etiologies [59,60,61].

Several clinical and paraclinical findings (tachycardia, persistent pain, pleural effusion, persistent leukocytosis, acidosis, and increased serum lactate) should raise the suspicion of complications, such as perforation [17]. The decision to perform emergency surgery must be heavily weighted due to increased morbimortality in the acute phase, as well as the possible negative impact on the long-term functional outcome [20]. Avoidance of major resection surgeries and conservative management are both recommended in the absence of clear evidence of tissue necrosis, perforation, or uncontrolled bleeding (Table 2).

The devastating effect that the ingestion of corrosive substances might have on the respiratory tract must also be mentioned. The aspiration of corrosive substances may induce extensive tracheobronchial injuries [63]. Thus, the role of bronchoscopy becomes essential in the presence of respiratory symptoms or when aspiration is suspected [64,65]. Complications such as airway necrosis, tracheobronchial perforation, aspiration pneumonitis, and the development of fistulas with the connecting mediastinal structures severely alter patient prognosis (Figure 2 and Figure 3) [5,64,65,66].

When the presence of extensive lesions in the upper digestive tract is suspected, the authors acknowledge the possible benefits of an initial bronchoscopy (prior to esophagogastroduodenoscopy) given that it provides warnings about the severity of oropharyngeal injuries and given that instruments with a smaller caliber are being used [5]. Moreover, bronchoscopy is a safe and repeatable procedure, both in the acute and chronic phases of intoxication.

Since one of the major consequences of tissue exposure to corrosive substances implies excessive fibrogenic activity, and since the secondary development of strictures is also a problem, additional therapies should be considered in selected cases. Several experimental corrosive esophagitis animal models have been used to promote new antifibrotic and antiproliferative strategies, such as vitamin E (α-tocopherol), methylprednisolone or EW-7197, and a selective inhibitor of TGF (transforming growth factor) β type I receptor kinase [67,68,69]. The administration of these agents in the early phases of the poisoning may limit the collagen deposition and stricture formation, but to date, they still lack clinical general acceptance.

Another major goal in the management of both the acute and chronic phases of corrosive substance poisoning is the avoidance of malnutrition [70]. Patients with grade 2B and 3A documented injuries require total parenteral nutrition for at least 6–8 weeks after the ingestion. Nutritional support might also be facilitated by the insertion of a feeding jejunostomy [18].

The accumulated evidence is consistent with the following recommendations assessing several controversial therapies in the management of acute exposure to corrosive substances (Table 3).

### 3.6. Interventional/Surgical Treatment

According to Katz and Kluger, over 90% of patients with grade 3 injuries and about 30–70% of patients with grade 2B will develop esophageal strictures within 8 weeks after oral exposure to corrosive substances [17]. A smaller number of cases complicated with gastric outlet obstruction have been described, with a very large distribution of the reported prevalence (5–60%) [17,18]. Current recommendations are a barium swallow study 4 weeks after ingestion in order to accurately evaluate the presence of esophageal or gastric outlet strictures, as well as to offer information on the location, length, number, or diameter of the strictures [18,72].

Dilatation of the esophagus is considered a relatively high-risk intervention, and determining the precise moment to perform endoscopic dilatation of the strictures therefore has a major impact on the functional outcome [17,72]. Complex strictures are considered to have a length >2 cm and a tortuous/angulated morphology, or they severely alter the luminal diameter [74]. The dilatation technique involves the use of a balloon or wire-guided bougie dilators, preferably under fluoroscopic guidance to enhance safety, and it requires multiple sessions to achieve optimal results [18,72,75]. The 2018 UK Guidelines on esophageal dilatations recommend a time interval of less than 2 weeks between the sessions [72].

Additional procedures, such as intralesional steroid injection or topical applications of mitomycin-C, can limit the synthesis of collagen and the excessive development of fibrosis, thus reducing the necessity for serial dilatation maneuvers [18]. The temporary placement of stents (optimum duration between 4 and 8 weeks) can represent an option in refractory strictures [72].

Late reconstructive surgery should be considered as a last-resort option for refractory or complicated complex strictures or for long strictures that are endoscopically inaccessible [17].

### 3.7. Disposition and Follow-Up

To date, corrosive-substance poisoning remains a therapeutic challenge. The most severe forms of poisoning can lead to mortality or long-term sequelae and can require multiple admissions and prolonged nutritional support [76].

During the long-term follow-up, endoscopic surveillance is recommended to allow early detection of pre-cancerous dysplastic lesions. No consensus has been established regarding the optimum interval to perform endoscopy, but every 2 to 3 years seems to be the most accepted strategy among specialists [51]. Nonetheless, given the features of corrosive-induced esophageal cancer (preexisting dysphagia due to the presence of chronic strictures and various morphologies of the luminal narrowing that may limit the accuracy of endoscopic mapping), additional paraclinical investigations, such as barium esophagography and/or contrast computed tomography imaging tests, may be needed to successfully detect malignant lesions [77].

### 3.8. Outcome

In the acute phase, the severity of corrosive poisoning may vary, from minor injuries to life-threatening conditions, such as perforation, bleeding, necrosis of the esophagus and stomach, sepsis, and shock [45,54,78,79]. The most common complications involve esophageal and gastric damage that may lead to an extensive scarring process, various degrees of luminal stricture, malnutrition, and a higher (more than a 1000-fold) risk of malignant transformation of the altered epithelium [51,80].

The estimated amount of time needed for esophageal cancer to develop is about 15-to-40 years after corrosive substance exposure and the initiation of the carcinogenesis process [51,81]. With a prevalence of up to 16% of cases, the predominant histological type of esophageal cancer is the squamous cell carcinoma [51,77,80,81]. The expansion of the tumor from a cicatricial tissue seems to have a favorable impact on the outcome of patients and the 5-year survival rate, since the intra- and peri-tumoral presence of fibrotic tissue might limit the local spread and lymphatic dissemination [51,80,82].

## 4. Conclusions

The management of post-corrosive esophageal lesions is a multidisciplinary one that must combat short-term complications, such as perforation and hemorrhage, but also long-term complications, such as stenosis and malignancy. Short-term complications are the prerogative of an interdisciplinary team that must act urgently to save the patient’s life by preventing perforation and hemorrhagic shock. Problems arise in the long-term management for treating the phenomena of fibrosis and stenosis characteristic of esophageal corrosive injuries. The first step is endoscopic dilation of the stenoses, which is followed by a surgical procedure if endoscopic dilation is impossible to perform or has a high risk of perforations. Esophageal stripping with different esophagoplasty techniques depending on the local lesions and the experience of the surgeon is the gold standard for the surgical treatment of post-corrosive esophageal strictures.

Although significant legislative limitations have been implemented, potentially corrosive substances are still easily accessed by the population, thus requiring supplementary measures to prevent corrosion-induced injuries. Ultimately, providing the best outcome for patients remains the main challenge.

## Figures and Tables

**Figure 1 jpm-13-00987-f001:**
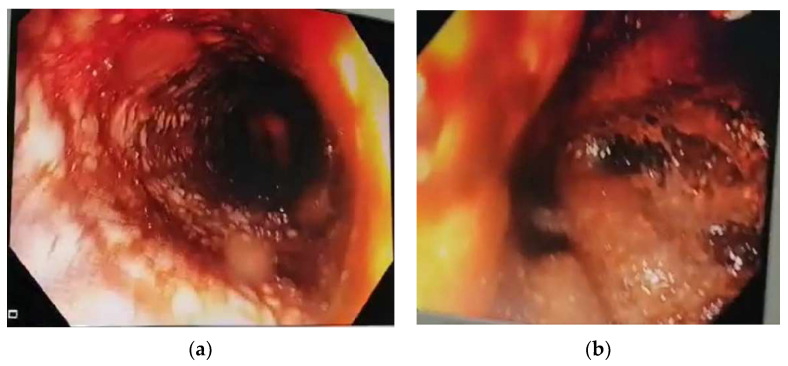
Endoscopic features upon admission (<12 h after ingestion of the corrosive substances). (**a**) Esophagoscopy: circumferential lesions, ulcerations, white exudates, mucosal erythema, and friability (grade 2B). (**b**) Gastroscopy: multiple deep ulcerations with black discoloration (grade 3A).

**Figure 2 jpm-13-00987-f002:**
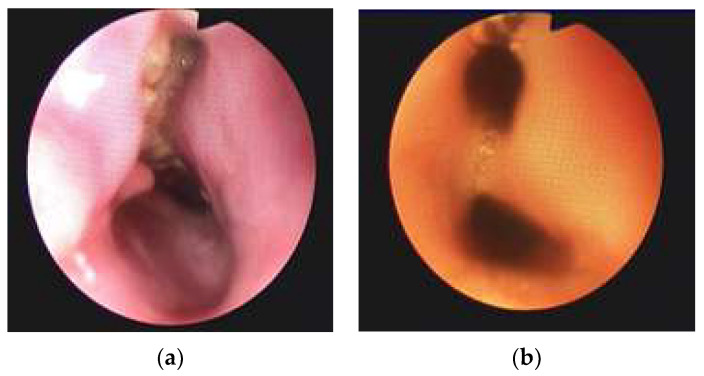
(**a**) Formation process of an esophageal–bronchial fistula. (**b**) Constituted esophageal-bronchial fistula 1 week after caustic soda ingestion (L. Sorodoc Collection) (reproduced from References [5,66], with permission).

**Figure 3 jpm-13-00987-f003:**
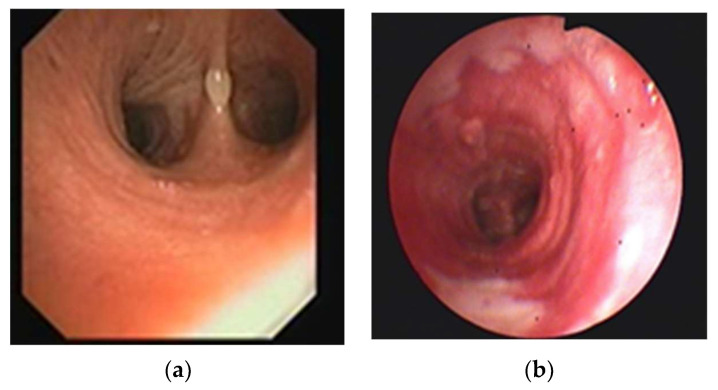
(**a**) Water blister at carina level after caustic soda ingestion; (**b**) inflammation and white membranes on the trachea after lye ingestion (O.R. Petris Collection) (reproduced from References [5,66], with permission).

**Table 1 jpm-13-00987-t001:** Endoscopic grading and prognosis of corrosive injuries (adapted from References [5,19,21,36,49]).

Injury Grading	Endoscopic Features	Prognosis
Grade 0	Normal	No sequelae
Grade 1	Mucosal oedema and mild erythema	Recovery **without** sequelae
Grade 2A	Superficial ulcerations, erosions, white exudates (patchy or linear), hemorrhages, mucosal erythema, and friability	Recovery **without** sequelae
Grade 2B	Circumferential lesions Deep ulcerations	Recovery **with** sequelae (strictures)
Grade 3A	Multiple transmural ulcerations and areas of necrosis with brownish-black or grayish discoloration. Complete intraluminal obliteration is possible	Recovery **with** sequelae (strictures)
Grade 3B	Extended necrosis	Recovery **with** sequelae
Grade 4	Perforations	Strictures in survivors

**Table 2 jpm-13-00987-t002:** Evidence-based guidelines for the updated management of corrosives ingestion injuries—acute phase (adapted from References [5,17,23,36,62]).

Pre-Hospital Management			
Patient with suspected corrosive ingestion	**Toxicological context**: Intentionality and determine substance, amount, and concentration route of poisoning; time of exposure; and co-ingestion of other toxic substances		
**MULTIDISCIPLINARY APPROACH** Emergency physician, internist–clinical toxicologist, gastroenterologist otorhinolaryngologist, intensive-care specialist, pulmonologist general surgeon, thoracic surgeon, psychiatrist, and radiologist			
**Critical/Unstable**	**Resuscitate in compliance with the Advanced Life Support guidelines**		
	Airway assessment	If there is laryngeal edema	Adrenaline nebulizers intravenous corticosteroids
		If there is airway deterioration	Definitive airway surgical tracheostomy/endotracheal intubation
	Cardio–circulatory stabilization	Fluid intravenous replacement; vasoactive agents Prevention of hemorrhagic shock	
**Respiratory and hemodynamic** **Stability**	Clinical or imagistic signs of perforation	**YES** (Grade IV): Emergency surgery	
		**NO: Early emergency esophagogastroduodenoscopy**––optimal within 12–24 h post-ingestion	
**The absence of symptoms does not correlate with the severity of the poisoning and does not rule out** **the necessity of performing an emergency endoscopy**			
**Moderate-to-severe injuries impose restriction from any oral intake**			
Grade 0–2A	Low risk of developing complications; progressive resumption of oral nutrition discharge when oral diet tolerated, and psychiatric evaluation if it was an attempted suicide No follow-up needed		
Grade 2B–3B	High-dose intravenous-inhibitor proton pump, total parenteral nutrition, or, preferably, insertion of jejunostomy tube for enteral nutrition symptomatic relief		

**Table 3 jpm-13-00987-t003:** Controversies concerning the management of acute corrosive poisoning (adapted from References [17,18,44,46,71,72,73]).

	Arguments	Counterarguments	Author’s Recommendation
Neutralization of the toxin	Traditional approach in the general population	Exothermic reactions and aggravation of tissue damage	Prohibited
Corticotherapy	Stricture prevention	Increased risk of perforations, bleeding, infections	Recommended in selected cases
Prophylactic antibiotherapy	Stricture prevention In association with corticosteroids In the case of confirmed perforation	Lack of evidence Risk factor for Clostridium difficile infection	Not recommended
Endoscopic placement of nasogastric tube	Maintains luminal integrity Reduces risk of stricture formation Facilitates enteral nutrition support	Cautions in the case of associated coagulation abnormalities Pathogenic colonization of the oropharynx, mainly related to Gram-negative bacteria Can facilitate the development of long strictures	Recommended in selected cases
Early bougienage (<3 weeks post ingestion)	For relief of stricture formation	High risk of perforation	NOT recommended
Stent to avoid stricture	Efficient in 52–72% of cases	High migration rate High endoscopy skills required for placement Increased procedure costs	Recommended in selected cases

## Data Availability

Not applicable.

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
