# Peer review of "Severe Intentional Corrosive (Nitric Acid) Acute Poisoning: A Case Report and Literature Review"

_jpm, 2023, doi:10.3390/jpm13060987_

Round 1

Reviewer 1 Report

Dear authors

I received your submission as a reviewer. I found that you tried to review management process of nitric acid poisoning via presenting a case presentation. That would be valuable, and I have some comments to help you improve the paper.

-      I suggest you revise the title as “Sever corrosive gastrointestinal injury following intentional ingestion of Nitric Acid: a Case Report and Literature Review”

-      According to the journal structure for case reports, the text of abstract must have continuity and you should remove the term “conclusion”.

-      I highly recommend you to use keywords compatible with MeSH system.

-      As you reviewed this type of poisoning in the discussion part, It is suitable to make a brief introduction to show the rarity, importance, and gap of knowledge regarding the topic, and replaced most of current text to the discussion part.

-      The second paragraph is more suitable to be replaced to the end of introduction to emphasize on what exactly you intend to focus on in this paper.

-      I highly recommend you to reorganized the discussion part and consider a proper order for presenting your literature review: pathophysiology, presentation, diagnosis, primary management, paraclinical valuation, secondary (management), disposition, outcome, etc. This would help the audience to easily and directly get pearl points from your article.

Kind Regards

Author Response

We sincerely appreciate the time and effort taken in reviewing our manuscript and providing us your feedback. Thank you for the insightful and valuable comments. We revised the manuscript and incorporated the suggestions made. Those changes are highlighted within the manuscript. Additional references were incorporated.

Best regards.

Reviewer 2 Report

I read with great interest your work. I think it is well written and length is acceptable. I have no further comments.
Best regards.

Author Response

We sincerely appreciate the time and effort taken in reviewing our manuscript and providing us your feedback. 

Best regards.

Reviewer 3 Report

Thank you for the opportunity to review this case report/review of the literature.

After reading the article I regret to say that it does not meet the formal and context-related requirements.

The number of authors is far too high for a case report. There are numerous language flaws and the message of the case, while being interesting, remains unspecific.

I don´t think that this article adds much to the literature but I would encourage the authors to edit the case according to the guidelines of a case-report journal.

Sorry for this direct statement of my personal opinion. Best regards

Extensive language and style editing required.

Author Response

We appreciate the time and effort you have put into your comments on our manuscript. Your objective reviews were most helpful and we sincerely hope that you'll find an improved version of the revised manuscript. 

As a team greatly involved in the field of Clinical Toxicology, our goal was to increase awareness on a very debilitating, yet preventable poisoning.

We consider that an integrated approach, involving both practical aspects and theoretical updates, will enhance a better understanding of the topic.

Although no significant difficulties arise from establishing a positive diagnosis, providing the best outcome for the patients represents the main challenge.

The complex presentations in the severe forms of corrosive substances poisonings often elude the standard recommendations regarding the best strategy to follow. Thus, a consensual procedure is still needed and to date, the relevant literature is inhomogeneous.

The selection of the authors reflects in fact the interdisciplinary management required for the severe forms of intoxication, the need for close follow up and repeated hospital admissions. All these aspects eventually translate into a heavy burden on the healthcare system, involving an increased number of resources (both human and financial). All mentioned authors directly participated in the management of the patient.

Round 2

Reviewer 3 Report

Thank you for the revised version of the manuscript. The authors reworked the paper extensively. The literature review character is now better visible and the structure is improved but the language and style has still serious flaws (e.g. present tense instead of past tense in the case report section). I would strongly insist on a professional editing. My previous suggestion to reduce the number of authors has also not been considered. Nineteen authors (including four "contributed equally") seem not appropriate to me, I would recommend not more than 4-5 autors.

Extensive language and style editing required.

Author Response

All recommended previous comments are greatly appreciated and helped us to refine our work. Professional English editing of the manuscript has been carried out (MDPI English editing ID: English-66890).

The previous suggestion to reduce the number of authors has been most attentively considered. Nevertheless, we must strongly admit the contribution of the involved physicians in the management of the patient. As presented in the manuscript, more than 10 months of follow-up was carried out, involving multiple admissions in several Departments (internal medicine, gastroenterology, general surgery, intensive care unit).

By reducing the number of authors, we would fail to acknowledge their medical involvement. From our point of view, this initiative is regrettably unsuitable.